# Pensions and Depressive Symptoms of Older Adults in China: The Mediating Role of Intergenerational Support

**DOI:** 10.3390/ijerph18073725

**Published:** 2021-04-02

**Authors:** Hui He, Ling Xu, Noelle Fields

**Affiliations:** 1School of Public Administration, The Xiangtan University, Xiangtan 411100, China; hehui@xtu.edu.cn; 2School of Social Work, The University of Texas at Arlington, Arlington, TX 76019, USA; noellefields@uta.edu

**Keywords:** CLASS, depressive symptoms, intergenerational support, pensions, older adults

## Abstract

This study aimed to investigate the relations between pensions and depressive symptoms of Chinese older people, and whether intergenerational support mediated such association. Secondary data was drawn from Chinese Longitudinal Aging Social Survey (CLASS) 2014 (*N* = 6687). Depressive symptoms were measured by 12-item version of the Centre for Epidemiological Studies Depression scale (CES-D). Intergenerational support was measured by financial, instrumental, and emotional support. About 80.1% of the participants had pension and the mean score of depressive symptoms of the participants was 17.10 (*SD* = 4.35) with a range from 12 to 36. The results from hierarchical linear regression revealed that there was significantly association between pensions and lower levels of depressive symptoms (*B* = −0.645, *p* < 0.000). Findings of mediation analyses also indicated that financial support from adult children played a mediating role between pensions and depressive symptoms (*B* = −0.039, 95% CI [−0.064, −0.018], *z* = −3.082, *p* = 0.002). Findings from this study enrich our theoretical and practical understanding of the roles of intergenerational support, and offer implications for social insurance policy, social work, and family support interventions for Chinese older adults.

## 1. Introduction

China is the most populous country in the world. The older adult population aged 60 years and above reached 253 million at the end of 2019, accounting for nearly 18 percent of the Chinese population, and 25 percent of the older adult population in the world [1]. China’s aging population will reach a peak which will account for nearly 30 percent of China’s population and one in six of the worldwide older adult population is Chinese by the middle of this century [2]. The rising number of older adults presents mental health issue challenges. For example, depression is a considerable public health problem that influenced the mental health of older adults worldwide [3,4]. An estimated 7% of the general older adults population suffered from unipolar depression and it accounted for 5.7 percent of the total disability (disability-adjusted life years) of older adults [5]. In China, approximately 31.2 percent of older adults manifest depression symptoms [6]. With the high prevalence rate of depressive symptoms in China, it is extremely urgent to face this challenge positively and help older adults to maintain mental health, which is also a significant factor to realize successful aging that is directly associated with the quality of life and well-being of the older adults [7].

Filial piety is a traditional Chinese cultural value that urges children to respect and take care of their parents [8]. Children’s intergenerational support is often an important guarantee for older adults to obtain subjective well-being, which may alleviate depression in older adults [9,10,11,12]. Study had shown that pensions affected children’s intergenerational support for their older parents [13]. As a steady source of income, empirical studies also showed that the pension system played an important role for older people to improve their mental health [14]. Therefore, theoretically, intergenerational support among older adults may play a mediating role in the relationships between pensions and depressive symptoms in China. While there are a few studies on the relationship between pensions and depressive symptoms among Chinese older adults [15,16], it remained unknown through what mechanisms that these associations may occur. This study investigated the relationship between pensions and depressive symptoms and examined whether and how intergenerational support mediated such association within the Chinese context where the pension system was implemented in recent years. Studies on this topic will provide healthcare and social service professionals with more information related to designing prevention and/or intervention programs that address depressive symptoms in the community.

### 1.1. Pension in China

As the main institutional arrangement to meet the fundamental living demands of older adults, pensions have shown to be an important way for the Chinese government to cope with the decline of traditional family functions because of modernization and industrialization [17]. The pension system in China includes three major parts: a basic pension system for urban workers (UW), the pension system of public institutions (PI) and a basic pension system for urban and rural residents (URR). At present, the Chinese pension system covers 67.8 percent of all residents [18], however, the social pensions in China may not fully meet the needs of older adults and alleviate the increasing financial burden of old adults, especially in rural areas [19,20]. Nevertheless, pensions determine the living standard of old adults and thus is considered as a primary security mechanism for the older adults in China [21], even though the amounts of pensions remain meager.

### 1.2. Pensions and Mental Health among Older Adults

Pensions are an important variable that is conductive to understanding geriatric depression. There is a potential risk of depressive symptoms that older adults are more likely to suffer from prolonged poverty than other groups [22]. It is particularly true for those in low-and middle-income countries [23]. Pensions are generally viewed as one of the sources of pooled income, which could protect the mental health of the older adults by securing their ability to obtain and utilize resources or decrease income inequality [24,25]. As an important formal social support, empirical studies have shown that the income from pensions plays an important role in their mental health across the world. For example, empirical studies conducted in Europe [26], Mexico [27], South Africa [28], South Korea [29] and Brazil [30] have found a positive association between pensions and mental health.

Studies on this topic among Chinese older adults have shown that pensions directly affected their mental health, such as reducing anxiety and improving life satisfaction [31]. Using China Health and Retirement Longitudinal Study (CHARLS), one study showed that pension income was directly and negatively related to depressive symptoms in rural China. Using the national sample of China Family Panel Studies (CFPS), another study found pension enrollment in rural China generated moderate to severe reductions in depressive symptoms, and such a beneficial effect was more pronounced among older adults than younger cohorts [32].

### 1.3. Pension and Intergenerational Support

#### 1.3.1. Theoretical Framework

How the public pension and system the support from children influence each other has often been in discussed with reference to the crowding-in versus crowding-out argument [33]. Crowding-out assumes that high public pension levels tend to reduce the need for support from children, whereas crowding-in implies that high pension levels tend to stimulate family help or support from children [34].

Two theoretical models can be used to explain these two contrary effects: the altruism model and the exchange model [35,36]. According to altruism model, the support or transfers from adult children to their parents are based on altruistic behavior, such as when their parents’ income is low. Thus, this model hypothesizes that public pensions will crowd out support from adult children to their parents. The exchange model assumes that transfer or support from adult children to their parents is contingent on various resources they receive from their parents at different stages of life. According to the exchange model, public pensions for older parents may crowd-in/increase or crowd-out/decrease the support or transfers from children, depending on the exchange levels between the two generations, either long-term or short-term exchange. For example, in terms of long-term exchange, adult children want to pay back the costs of education and/or child-raising expenditure provided by parents in and thus provide support to aging parents regardless of whether their parents have a pension (crowding-in effect). In terms of short-term exchange, if parents expect a higher income because of a pension, they may spend more money or provide more help to adult children, which may later result in higher support from their adult children as a result of exchange (crowding-in effect). However, older parents may still have limited disposable incomes because of the low amount of average pensions, and thus may provide little financial or other support to their adult children, which in turn could result in little or no transfer or support from their children (crowding-out effect).

#### 1.3.2. Empirical Studies

Studies on the effects of pensions on support or transfer from children indicate both crowding-out and crowding-in effects. On the one hand, economists have long suggested that higher pension benefits crowd out other sources of household wealth accumulation. It was found that each dollar of pension wealth is associated with a 53–67 cent decline in non-pension wealth in the USA [37]. Studies reported that the increase of public transfers significantly crowded out private financial transfers from children to parents in South Korean [38], Taiwan [39] and Hong Kong [40]. For rural Chinese older adults, research found that to a certain extent, pensions reduced their dependence on adult children and crowded out the time and support provided by adult children [41,42]. Studies also showed that the implementation of the pension system in China alleviated children’s burden of providing support to aging parents in rural China.

On the other hand, studies found that pensions have a crowding-in effect on children’s intergenerational financial support [43,44,45,46]. Pensions increased and improved the disposable resources and affordability of older parents, expanded intergenerational support within the family, and eventually presented a crowding-in effect on support from children. Similarly, a study in Norway, England, Germany, Spain, and Israel also found the welfare state has not crowded out the family from elder care, but has rather helped the generations establish more independent relationships, which suggests strong emotional support and/or generational bonds.

### 1.4. Intergenerational Support and Depressive Symptoms

Intergenerational support from children plays an important role worldwide in supporting the mental health and well-being of older parents throughout their lives. For example, studies showed that emotional support from adult children had a positive role in promoting the health of the older adults, which greatly improved their subjective well-being [47,48]. For instrumental support, research found that the adequacy of instrumental support was associated with a lower likelihood of being categorized as depressed among community-dwelling older women with a disability in the USA [49]. Research in China also found that depressive symptoms were often reduced by assistance from daughters-in-law and sometimes increased when such support was from sons in rural China [50]. In terms of financial support, studies have shown that older adults receiving financial support and daily care from their children have a positive effect on their physical and mental health in South Korean [51] and Eastern European [52]. Studies in China also showed that the financial support provided by children met the physical and economic needs of older adults in rural areas and helped to elevate their mental health [53]. Finally, research suggested that the more financial support received from adult children, the lower the level of depressive symptoms for both rural and urban Chinese older adults.

### 1.5. The Present Study

The above literature showed the potential associations between pensions and intergenerational support, as well as intergenerational support and depressive symptoms. These pathways suggest that intergenerational support may theoretically play a mediating role in the relationship between pensions and depressive symptoms. However, no previous studies have investigated such mechanism of intergenerational support. Having a better understanding of the pathway or mechanism of how pension associated depressive symptoms may provide guidance for improving the delivery of appropriate programs, services, or interventions to promote the psychological well-being in older adults.

To address the abovementioned gaps in the literature, the present study attempted to examine the relationship between pensions and depressive symptoms of Chinese older adults and tested whether intergenerational support mediated such association. Specifically, we aimed to investigate the following two research questions and associated hypotheses.

Are pensions associated with depressive symptoms among Chinese older adults?

**Hypothesis** **1** **(H1).**
*We hypothesized that receiving a pension was negatively associated with depressive symptoms. The more pensions the older adults received, the lower the possibility of depression could be.*


How did intergenerational support (emotional, instrumental, and financial support from children) mediate such association?

**Hypothesis** **2** **(H2).**
*The hypotheses were: receiving a pension resulted in higher levels of instrumental support (H2.1), financial support (H2.2) and emotional support (H2.3) because of the filial piety, which in turn led to fewer depressive symptoms.*


## 2. Materials and Methods

### 2.1. Study Design

Quantitative research design was used in this study. Pensions, intergenerational support, and depressive symptoms were the key variables of the study. Depressive symptoms were the dependent variable, pensions were the independent variable, intergenerational support was the mediator, and several demographic variables were used as control variables. This study was designed to answer whether pensions were negatively related to depressive symptoms among Chinese older adults, and whether intergenerational support from adult children played a mediating role between pensions and depressive symptoms. The conceptual framework is shown in Figure 1. The changes of pensions affect the intergenerational support of children (including emotional support, instrumental support, financial support) (path a), while intergenerational support of children also affect the depression symptoms of the older adults (path b), and pensions affect the depression symptoms of the older adults (path c). In addition, under the mediators, pensions affect depression symptoms of the older adults (path c’). We used a publically available dataset and conducted secondary data analysis, therefore no informed consent for the participants was needed nor ethical approval from the Institutional Review Board.

### 2.2. Data and Sampling

The study utilized data from the 2014 Chinese Longitudinal Aging Social Survey (CLASS), a nationwide, continuous large-scale social survey project. This data was collected by Renmin University of China in conjunction with several academic institutions all over the country. The CLASS survey was conducted with a stratified multistage probability sampling method. First, the county areas were selected to act as respondents (including the counties, the county-level cities or districts). Next, the village committees in rural areas or the residential committees in urban areas were selected as the secondary sampling unit. Third, the people aged over 60 years were selected as the survey respondents. The survey covered 29 provinces, autonomous regions, or municipalities, including 462 village committees or the residential committees. In every village committee or residential committee, the graphic sampling method was adopted to capture the family household unit with only one older person being investigated within each family unit. Several survey teams with a total of more than 800 staff were trained and employed to conduct face-to-face interviews. To ensure the high quality of the survey, the CLASS adopted on-site and remote data checking strategies. More specifically, the researchers recruited about 30 supervisors who then spent nearly 1.5 months (on average) to visit all of the sites to carry out the on-site quality supervision and control. For the remote data checking approach, 30% of the questionnaires were randomly selected in each site for a telephone check and an on-site re-check was offered if any problems were found in the survey. After deleting the participants who failed the cognitive ability test and those who did not have any children, the final sample size for this study was 6687.

### 2.3. Measurement

Key variables in this study included pension, intergenerational support, and depressive symptoms.

#### 2.3.1. Dependent Variable

Levels of depressive symptoms were measured with the Depression Scale (Center for Epidemiological Survey, CES-D) by asking the participants how they felt during the past week. Twelve questions were asked, such as “did you feel in a good mood in the past week”, “were you sleepless in the past week”, “did you have someone to company you in the past week”, etc. Items were rated on 3-point scale (1 = *no* to 3 = *very often)*. Cronbach’s alpha for this scale was 0.761. After reversely recording three positive items, sum scores were calculated with a range of 12 to 36. The higher scores indicated higher level of depressive symptoms.

#### 2.3.2. Independent variables

Pensions were measured by asking the question “what kind of basic pension have you been provided/offered?” The insurance choices included “UW, PI, URR and new rural pension (a part of old form of URR still carried out in some areas of China).” Participants who responded that they had any one of the above four types of insurance were defined as “1 = *yes*”. Participants who choose none of the four insurances was defined as “0 = *no*”.

#### 2.3.3. Mediators

As the core of social support for older adults in China, intergenerational support is conceptualized in terms of emotional support, instrumental support, and financial support [54]. In the present study, intergenerational support was similarly measured using emotional support, instrumental support, and financial support. Two questions were selected to measure the emotional support. One was “From every sides of consideration, how close you are with your children?” (1 = *no close* to 3 = *close*). The other one was “Do you think that this offspring does not give you enough care?” (4 = *never* to 1 = *often*). The answers were added as a continuous variable to measure the emotional support for final analysis [55]. Instrumental support was measured by the question “How often has this child helped you do housework in the past 12 months?” (1 = *none* to 5 = *almost every day*). Finally, financial support was measured by the following question “In the past 12 months, has this child ever given you (or your living spouse) money, food or gifts, and how much is the total value of these goods?” (1 = *none* to 9 = *more than RMB12,000*, which is close to $1745). The above questions were collected by asking the older adult participants to report/respond for each of their children (up to five children). Maximum values for each support were used if the participants had multiple children.

#### 2.3.4. Control Variables

This study selected several demographic questions pertaining to the older adults as control variables [56], including age in years, gender (1 = *male*, 0 = *female*), living area (1 = *urban*, 0 = *rural*), whether they had a chronic illness (1 = *yes*, 0 = *no*), marital status (1 = *married*, 0 = *not married,* including widowed, divorce, and unmarried), income (1= *have engaged paid word or activities*, 0 = *no payment from work*), whether living with others (1 = *yes*, 0 = *no*), and social networks. The Lubben Social Network Scale (LSNS-6) [57] was used to measure social networks. The LSNS-6 is a validated instrument designed to gauge social isolation in older adults by measuring the number and frequency of social contacts with friends and family members. Six questions were asked for each participant, such as “How many friends/family members can you turn to when you need help?”, “How many friends/family members do you meet or get in touch with at least once a month?” etc. Items were rated on 6-point scale (*0 = none to 5 = nine and above*). Cronbach’s alpha for this range was 0.761. The LSNS-6 total score is an equally weighted sum of these six items with a range from 0 to 30. The higher scores indicated higher level of social networks.

### 2.4. Data Analysis

Before running regressions, descriptive statistical analysis and frequencies of variables were conducted to describe participants’ demographic backgrounds. Multicollinearity among independent variables was also tested before the regression. The results indicated that the variance inflation factor (VIF) was less than 2.5 and the tolerance value was greater than 0.1 [58], which showed that there was no multicollinearity between the independent variables. Bivariate correlation analyses were also conducted among key variables. The above analyses were conducted using SPSS 22.0. Finally, multivariate regression analyses were conducted to examine the mediation effects of intergenerational support by PROCESS for 3.0 [59]. This study controlled for age, gender, living area, marital status, living with others, having income, chronic illness, and social networks because previous study showed they were significant associates of depression [60,61]. We used listwise deletion to address missing data because less than 5% of them are missing data and they are missing completely at random [62].

## 3. Results

### 3.1. Sample Description

The demographic characteristics of the participants are provided in Table 1. The average of age was 68.96 years old (*SD* = 7.44). More than half of the participants were males (55.2%) and about 67.3% were living in urban areas. About 87.5 percent were married and only 19.4% were engaged in paid work or activities. About 70.8% of the participants had a chronic illness. The mean social network score was 15.40 (*SD* = 6.45), with a range from 0 to 30. About 80.1% of the participants had pension. Generally speaking, the mean score of depressive symptoms of the participants was 17.10 (*SD* = 4.35), with a range from 12 to 36. Among the three indicators of intergenerational support, the mean scores of emotional, instrumental, and financial supports were 4.67 (*SD* = 0.83), 3.03 (*SD* = 1.64) and 4.62 (*SD* = 2.15), respectively.

### 3.2. Bivariate Relationships among Key Variables

Bivariate correlations were used to test the associations between the key variables (see Table 2). Pension (*r* = −0.085, *p* < 0.01), emotional support (*r* = −0.162, *p* < 0.01), instrumental support (*r* = −0.031, *p* < 0.01) and financial support (*r* = −0.128, *p* < 0.01) were all significantly and negatively related to higher levels of depression. Additionally, pension (*r* = 0.053, *p* < 0.01) was significantly related to financial support, but not with instrumental or emotional support from children.

### 3.3. Pensions, Intergenerational Support and Depressive Symptoms

Three independent regression models were conducted to examine the mediation effects of intergenerational support (emotional, instrumental, and financial support) on the associations between pensions and depressive symptoms of Chinese older adults.

As shown in Table 3, Model 1 (path c) showed a significant relationship between pensions and depressive symptoms after controlling for all the demographic variables. Results indicated that pensions (*B* = −0.645, *p* < 0.000) were significantly and negatively related to the depressive symptoms of older adults. Model 2 (Path a1) was about the association between pensions and emotional support, and it manifested there was no significant correlation between pensions and emotional support. Model 2 (path a2) showed a significant negative correlation between pensions and instrumental support (*B =* −0.131, *p* < 0.05). Model 2 (path a3) indicated that pensions had a significant and positive association on financial support (*B* = 0.237, *p* < 0.001). Model 3 (path b and c’) showed the relationship among pensions, intergenerational support and depressive symptoms of the older adults. The results showed that both emotional support (*B* = −0.682, *p* < 0.001) and financial support *(B* = −0.165, *p* < 0.001) were negatively correlated with depressive symptoms, while instrumental support had no significant correlation with depressive symptoms. Pensions (*B* = −0.534, *p* < 0.000) were still significantly and negatively associated with depressive symptoms after adding emotional, instrumental, and financial support, but the size decreased compared to its original association in path c (*B* = −0.645, *p* < 0.000). By using the PROCESS feature of SPSS [63,64], the indirect effect results showed that only financial support mediated the association between pension and depression (*B* = −0.039, 95% CI [−0.064, −0.018], *z* = −3.082, *p* = 0.002).

## 4. Discussion

Using nationally representative data from the 2014 CLASS survey, this study tested the relationship between pensions and depressive symptoms among Chinese older adults, as well as whether and how intergenerational support (emotional, instrumental, and financial support from children) mediated such an association. Findings suggested that pension was significantly associated with depressive symptoms, and only financial support mediated such an association.

### 4.1. Pension and Depressive Symptoms

This research found that pensions had a significant negative association with depressive symptoms of Chinese older adults. Therefore, Hypothesis 1 was supported. This finding was consistent with previous research indicating that pensions, as a type of stable source of income, had a positive effect on the mental health, including depressive symptoms, of older adults across the world. The study findings also confirmed that participating in the pension system significantly reduced the depressive symptoms of Chinese older adults as shown in the previous literature [65]. The beneficial effects of pension on depressive symptoms might be due to increasing fixed income and reduced economic uncertainty [66].

### 4.2. Pensions, Intergenerational Support, and Depressive Symptoms

This study also found that adult children’s financial support mediated the relationship between pensions and depressive symptoms among Chinese older adults. Therefore, Hypothesis 2.3 was supported. On the one hand, receiving a pension had a significant positive effect on financial support from children and thus confirmed the crowding-in effect reported in the literature. One potential reason might be that even though all Chinese older adults receive pensions when they age, however, the pension amount is very low or limited, especially in rural China [67]. They therefore might have difficulties in affording some living or medical costs. If there is a financial gap in the pension, the children will fill it accordingly [68]. Therefore, they still receive the financial support from their children [69]. The other reason is the relatively high pension income in urban China (compared to rural region) can improve the ability of urban older adults to financially support children or grandchildren [70]. The more resources that they can control, the wider the scope they will exchange, which leads to a crowding-in effect. Pensions may have enabled the older adults to support the education, training and health expenditure of the younger generation (i.e., the grandchildren), so the children in turn may have been willing to provide the aging parent more respect and care, thereby increasing the financial support given to their parents [71]. On the other hand, children‘s financial support to their parents was associated with lower levels of depressive symptoms, which was consistent with the literature notion that financial support from children can boost the mental health of the older adults in China as well as in other countries [72,73]. Therefore, pensions may have indirectly alleviated the depression of the older adults through their children’s financial support.

This study did not find that adult children’s emotional or instrumental support had a mediated relationship between pensions and depressive symptom and thus hypotheses 2.1 and 2.2 were not supported. This is notable, as filial piety is a traditional cultural value that urges children to respect and care for their parents in China. As a result, the receipt of pensions did not affect the emotional support from the adult children, which thus did not play a mediator role in the relationship between pensions and depressive symptoms. On the other hand, older adults having pensions to purchase the social services may lower the necessity of instrumental support from their children as shown by their negative association in the present study, which also coincide with a previous study [74]. However, the path from instrumental support to depressive symptoms was not significant in the present study, which was consistent with a previous study indicating that acquiring instrumental support from children was not related to parents’ level of depression in China [75]. When too much help is provided by the children, some older adults may feel guilty [76], which may add the possibilities of distress and depressive symptoms for the older parents [77,78]. Therefore, instrumental support did not have a mediating role between pensions and depressive symptoms.

### 4.3. Study Limitations

Our study had limitations that must be considered when examining the results. Firstly, this study mainly examined social pensions. However, as older adults age, their physical functioning often becomes a more significant concern for them. Thus, medical payments should also become a key aspect of any pension. However, we did not have such measures available in our data. Secondly, the cross-sectional data limitation cannot capture the true causal relationship and track the changes of relationship between pensions and depressive symptoms of the older adults over time. Future studies may consider to use longitudinal data to overcome such limitations. Finally, the measurement of intergenerational support is relatively crude. Some of them were measured by single or two items. A more comprehensive measurement or scales should be used to capture the different dimensions of intergenerational support received from children.

### 4.4. Implications

In spite of these study limitations, our study has important implications for care and/or social service providers and for the improvement of the pension policy for older adults in China. The study results indicated that pensions affected the mental health of older adults largely through intergenerational support. This highlights the importance of intergenerational support as it cannot be replaced by a formal system. There should be a complementary effect rather than a substitution effect between the family security and the social security system. The government can continue to encourage the younger generations to consciously practice filial piety and provide financial support to their parents [79,80] for the basic needs of life, such as clothes, food, housing, transportation, medical care, etc.

The findings of a negative association between receiving a pension and mental health should also be given attention. Such findings suggest a need to enhance the pension system and boost the level of pensions for older Chinese adults, especially for those with low-incomes and/or for those living in rural China. The government can improve the policies and regulations on such programs, such as ensuring the payment of the pension to the older adults on time and/or improving the level of the pensions year by year to nullify the negative effects of inflation, etc.

## 5. Conclusions

The results of the present study revealed that pensions were significantly negatively related to depressive symptoms, and intergenerational support (i.e., financial support) served as a mediator mechanism through which such associations occurred. The findings of the present study shed light on the significance of pensions and intergenerational support in the mental health of old adults in China. The study stresses the need to enhance the pension system for the health of older adults and reinforces a call for stronger assessments targeting their psychological needs. There is also an emergent need to recognize and deal with the traditional family function, particularly the financial support from children, in order to optimize the pension policies offered by the government.

## Figures and Tables

**Figure 1 ijerph-18-03725-f001:**
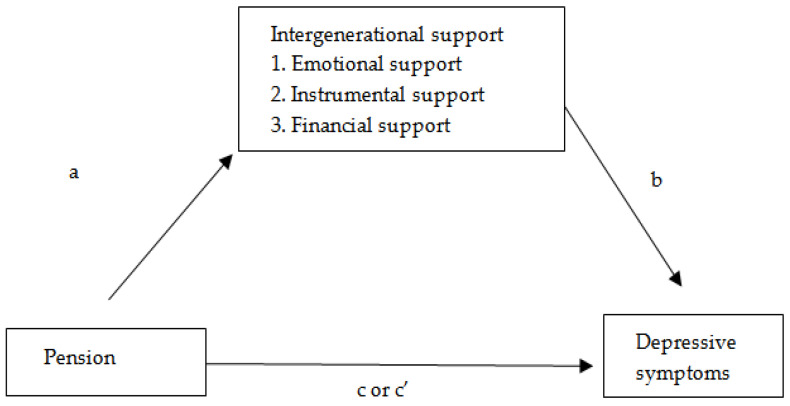
Conceptual framework of this study.

**Table 1 ijerph-18-03725-t001:** Sample Characteristics (*N =* 6687).

Variables	*N* (%)	Mean (SD)	%	Range
Age	0	68.96 (7.44)		60–113
Gender	0			
Male			55.2	
Female			44.8	
Living area	0			
Urban			67.3	
Rural			32.7	
Marital status (being married)	0.1		72.5	
Live with others (yes)	0.5		87.5	
Having income (yes)	0.1		19.4	
Chronic illness (yes)	0.8		70.8	
Social networks	17.7	15.40 (6.45)		0–30
Emotional support	1.0	4.67 (0.83)		1–5
Instrumental support	0.1	3.03 (1.64)		1–5
Financial support	4.1	4.62 (2.15)		1–9
Pension	1.8		80.1	
Depressive symptoms	6.2	17.10 (4.35)		12–36

**Table 2 ijerph-18-03725-t002:** Correlations among key variables (*N =* 6687).

Variables	1	2	3	4	5
1. Pension	1				
2. Depressive symptoms	−0.085 **	1			
3. Emotional support	0.011	−0.162 **	1		
4. Instrumental support	0.020	−0.031 **	0.136 **	1	
5. Financial support	0.053 **	−0.128 **	0.087 **	0.070 **	1

** *p* < 0.01.

**Table 3 ijerph-18-03725-t003:** Mediating effects of IS in the association between pension insurance and depressive symptoms (*N = 6687*).

Variables	Model 1		Model 2						Model 3	
	(c)		(a1_ES)		(a2_IS)		(a3_FS)		(b & c′)	
	B	SE	B	SE	B	SE	B	SE	B	SE
Age	−0.011	0.007	0.005 ***	0.002	0.021 ***	0.003	−0.016 ***	0.004	−0.010	0.008
Gender (ref = female)	−0.083	0.103	−0.069 **	0.022	−0.016	0.041	−0.050	0.055	−0.193	0.105
Living area (ref = rural)	−1.255 ***	0.110	−0.084 ***	0.023	0.019	0.043	0.227 ***	0.058	−1.293 ***	0.111
Marital status	−1.227 ***	0.135	0.021	0.028	−0.840 ***	0.053	0.001	0.072	−1.177 ***	0.139
Income (yes)	−0.604 ***	0.135	0.037	0.028	−0.075	0.053	−0.301 ***	0.072	−0.612 ***	0.136
Chronic illness (yes)	1.539 ***	0.109	−0.076 **	0.023	0.047	0.043	0.045	0.059	1.489 ***	0.111
Social networks	−0.137 ***	0.008	0.014 ***	0.002	0.034 ***	0.003	0.034 ***	0.004	−0.122 ***	0.008
Live with others	−1.21 ***	0.171	0.088 **	0.036	1.194 ***	0.068	0.151	0.092	−1.136 ***	0.178
Pension	−0.645 ***	0.129	0.034	0.027	−0.131 *	0.050	0.237 ***	0.068	−0.534 ***	0.130
Emotional support									−0.682 ***	0.060
Instrumental support									−0.002	0.032
Financial support									−0.165 ***	0.024
Constant	22.374 ***		4.121 ***		0.694 ***		4.791 ***		25.937 ***	
Adjusted R^2^	0.143		0.019		0.095		0.024		0.167	

* *p* < 0.05, ** *p* < 0.01, *** *p* < 0.001. Notes: Independent variable; pension; Dependent variable; depressive symptoms; In model 1, pension is the predictor, depressive symptoms is the outcome; In model 2, pension is the predictor and intergenerational support is the outcome (a1 for emotional support (ES), a2 for instrumental support (IS), and a3 for financial support (FS)); In model 3, pension is the predictor, intergenerational support is the mediator, and depressive symptoms is the outcome.

## Data Availability

Publicly available datasets were analyzed in this study. This data can be found here: http://class.ruc.edu.cn/index.htm (accessed on 2 February 2021).

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
