# Peer review of "Pensions and Depressive Symptoms of Older Adults in China: The Mediating Role of Intergenerational Support"

_ijerph, 2021, doi:10.3390/ijerph18073725_

Round 1
Reviewer 1 Report
Thank you for the opportunity to review the study "Pension and Depressive Symptoms of Older Adults in China: The Mediating Role of Intergenerational Support".
I have reviewed your paper following "Strengthening the Reporting of Observational Studies in Epidemiology (STROBE) statement". Please find the attached file. Basically, most items does not meet the criteria in the STROBE statement.

Author Response
Thank you so much for sending us the thoughtful comments. We have incorporated your comments in the revised manuscript. As you will see, the manuscript has been rewritten in line with these comments and we believe it is significantly strengthened. We have highlighted our major changes in the revised manuscript in red color when possible. The following is a point-to-point response to your comments.
Point 1: Title and abstract----There is no study design in your title. The result should be structured (e.g. IMRAD) and shown the statistical data (e.g. correlation coefficient, pvalue, 95% IC). No need to write about statistical software.
Response 1: Thank you for your comments about the tittle and abstract of the study. After carefully reading the title, we thought the current title is the most concise one that best represent the current study. Therefore, we did not change our title. For the abstract, we added relevant numbers and deleted statistical software as suggested (see red highlights in abstract).
Point 2: Introduction----Background/rationale. but a little over informative on pages 2-3. Please consider expressing simply.
Response 2: Thank you for your conductive suggestion. We have made some deletions in part 3.1. Furthermore, we simplified some long paragraph for greater clarity, and revised other sections.
Point 3: Introduction----Objectives. Please consider writing the hypotheses in a bullet.
Response 3: Thank you pointing this out. We restated the research questions and hypotheses using bullet points as well as H1, H2.1, H2.2, and H2.3 for greater clarity.
Point 4: Methods----Study design. You should write a study design section in your method at first. NOTE: There is no information about study ethics in your manuscript.
Response 4: Thank you for your comments. Our study design was described in part 2.1. This manuscript used publically available dataset and conducted secondary data analysis. So, there are no informed consent for the participants and no ethical approval from the Institutional Review Board.
Point 5: Methods----Setting. There is any information about the date.
Response 5: Thank you for your comment. The study utilized data from the 2014 Chinese Longitudinal Aging Social Survey (CLASS), a nationwide large-scale social survey project. We mentioned this in part 2.2.
Point 6: Methods----Participants. Please consider writing the eligibility criteria for participants and areas.
Response 6: Thank you for your comment. Older adults aged over 60 years were eligible for participating in CLASS. This study deleted the participants who did not have any children and those who failed the cognitive ability test. We mentioned this in part 2.2.
Point 7: Methods----Data sources/Measurement. Please show the citation for The Lubben Social Network Scale (LSNS-6).
Response 7: Thank you for your comment. We have provided citation as well as the reference in the revised manuscript. The following was: Lubben, J. (1988). Assessing social networks among elderly populations. Family & Community Health, 11(3), 42–52. doi:10.1097/00003727-198811000-00008
Point 8: Methods----Bias. There is no information about bias.
Response 8: Thank you for your comment. We do not think there is bias involved in this secondary data analysis paper.
Point 9: Methods----Study size. There is no information about sample size.
Response 9: The final sample size for this study was 6,687 after deleting the participants who did not have any children and those who failed the cognitive ability test (see part 2.2).
Point 10: Methods----Quantitative variables. Explain how quantitative variables were handled in the analyses. If applicable, describe which groupings were chosen and why. You should explain this in data analysis section.
Response 10: Thank you for your comments. How quantitative variables were handled were described in part 2.3.
Point 11: Methods----Statistical methods. You should explain all statistical methods, including those used to control for confounding in data analysis section. You should explain how missing data were addressed. If applicable, describe analytical methods taking account of sampling strategy. Please consider investigating any sensitivity analysis.
Response 11: This study controlled for age, gender, living area, marital status, living with others, having income, chronic illness, and social networks because previous study showed they were significant associates of depression (Choi & Schoeni,2017; Simpson, Albani, Bell, Bambra, & Brown, 2021). We used listwise deletion to address missing data because less than 5% of them were missing data and they were missing completely at random (Schafer, 1999). We added above information in part 2.4 (see red highlights). Because we did not use multiple imputation for missing data, we did not conduct sensitivity analysis.
Point 12: Results----Participants. You should explain report numbers of individuals at each stage of study—eg numbers potentially eligible, examined for eligibility, confirmed eligible, included in the study, completing follow-up, and analysed using flow diagram.
Response 12: Thank you for your comment. We agree with you the importance of detailed sampling procedure is important. Unfortunately, there was no such detailed information about how many participants were selected at each stage because we used second-hand data to analysis this study.
Point 13: Results----Descriptive data. Give characteristics of study participants (eg demographic, clinical, social) and information on exposures and potential confounders, but please consider to the item in the table 1 (e.g. mean, range and %). You should explain number of participants with missing data for each variable of interest using flow diagram.
Response 13: Thank you for your comments. The missing data in our sample is not an issue. We added another column of “N” in table one to report the frequency of each variable. From there, readers can have idea of the missing data for each variable.
Point 14: Results----Main results. But it is easy for reader to understand if you show the data (β, p value, and 95% CI) with the figure of Model.
Response 14: Thanks for the thoughtful comments. Since we have three possible mediators paths, if we show the data (β, p value, and 95% CI) in figure 1, it would be messy and readers would get confusions.
Reviewer 2 Report
Dear authors,
Thank you for giving me an opportunity to review your manuscript. I read it with interest and found high quality of methods and presentations. Your study highlighted the mediating role of intergenerational support in the relationship of pension and depressive symptoms in Chinese older adults.
Minor comments:
1) 2.2. Measurement: Describe the validity (sensitivity and specificity) of Chinese version of CES-D.
2) 3.3. Pension, intergenerational support and depressive symptoms: In the last sentence, you showed the mediational effect of financial support. Explain how to derive the regression coefficient (B) of -0.039.
3) Table 2: Need footnotes of "*" and "**", as *p<.05 and **p<.01.
4) Typographical errors:
5th line of page 3: Change "crowing-in" to "crowding-in".
2nd line from the bottom of page 6: Change "average age of" to "average of age".
Column headings of Table 1: Change ”Range” to "%" and "%" to "Range".
5) As authors mentioned in 1.1. Pension in China, the social pension could not fully meet the needs of older adults especially in rural areas. As shown in Table 3, living rural area was a significant factor of depressive symptoms. Some moderation model including living area as a moderator between pension and depressive symptoms would be helpful to reveal relationship between pension and depressive symptoms.
Author Response
Thank you so much for sending us the thoughtful comments. We have incorporated your comments in the revised manuscript. As you will see, the manuscript has been rewritten in line with these comments and we believe it is significantly strengthened. We have highlighted our major changes in the revised manuscript in red color when possible. The following is a point-to-point response to your comments.
Point 1: 2.2. Measurement: Describe the validity (sensitivity and specificity) of Chinese version of CES-D.
Response 1: Thanks for your suggestion. Because this study used second-hand data, the validity (sensitivity and specificity) of Chinese version of CES-D cannot be calculated.
Point 2: 3.3. Pension, intergenerational support and depressive symptoms: In the last sentence, you showed the mediational effect of financial support. Explain how to derive the regression coefficient (B) of -0.039.
Response 2: Thanks for your question. We obtained the coefficient of the indirect effect from the PROCESS output. In general,indirect effect = total (path C)–direct (path C’).
Point 3: Table 2: Need footnotes of "*" and "**", as *p<.05 and **p<.01.
Response 3: Thank you for reminding us of the footnotes of the table 2. All the significance level in table 2 were at p<.01. We added the footnote “ ** p <. 01” under the table 2.
Point 4: Typographical errors: 5th line of page 3: Change "crowing-in" to "crowding-in". 2nd line from the bottom of page 6: Change "average age of" to "average of age". Column headings of Table 1: Change ”Range” to "%" and "%" to "Range".
Response 4: Thanks for pointing out these typos, which is highly appreciated. We are very sorry for our typographical errors. We have carefully scrutinized the manuscript, and made corresponding revisions.
Point 5: As authors mentioned in 1.1. Pension in China, the social pension could not fully meet the needs of older adults especially in rural areas. As shown in Table 3, living rural area was a significant factor of depressive symptoms. Some moderation model including living area as a moderator between pension and depressive symptoms would be helpful to reveal relationship between pension and depressive symptoms.
Response 5: Thank you very much for your comments on the study. We will take into consideration of testing moderating effect of urban/rural area in our future research.
